# Novel Pyrroloquinoline Quinone-Modified Cerium Oxide Nanoparticles and Their Selective Cytotoxicity Under X-Ray Irradiation

**DOI:** 10.3390/antiox13121445

**Published:** 2024-11-24

**Authors:** Elizaveta A. Zamyatina, Olga A. Goryacheva, Anton L. Popov, Nelli R. Popova

**Affiliations:** 1Isotope Research Laboratory, Institute of Theoretical and Experimental Biophysics of the Russian Academy of Sciences, Pushchino 142290, Russia; zamyatinaea@iteb.pushchino.ru; 2Chemistry Institute, Saratov State University Named after N.G. Chernyshevsky, Saratov 410012, Russia; olga.goryacheva.93@mail.ru; 3Theranostics and Nuclear Medicine Laboratory, Institute of Theoretical and Experimental Biophysics of the Russian Academy of Sciences, Pushchino 142290, Russia; popoval@iteb.pushchino.ru

**Keywords:** cerium oxide nanoparticles, PQQ, X-ray, ROS, DNA damage, glutathione, mitochondria, fibroblasts cells, adenocarcinoma cells

## Abstract

Ionizing radiation leads to the development of oxidative stress and damage to biologically important macromolecules (DNA, mitochondria, etc.), which in turn lead to cell death. In the case of radiotherapy, both cancer cells and normal cells are damaged. In this regard, the development of new selective antioxidants is relevant. In this study, we first investigated the redox activity of cerium oxide-pyrroloquinoline quinone nanoparticles (CeO_2_@PQQ NPs) and their cytotoxic effects on normal (mouse fibroblasts, L929) and cancer (mouse adenocarcinoma, EMT6/P) cell cultures. Furthermore, the biological activity of CeO_2_@PQQ NPs was evaluated in comparison with that of CeO_2_ NPs and PQQ. The nanoparticles demonstrated pH-dependent reductions in the content of hydrogen peroxide after X-ray exposure. Our findings indicate that viability of EMT6/P cells was more adversely affected by CeO_2_@PQQ NPs at lower concentrations (0.1 μM) compared to L929. Following X-ray irradiation at a dose of 5 Gy, significant changes in mitochondrial potential (by 29%) and decreased glutathione levels (by 32%) were also observed in EMT6/P culture following irradiation and incubation with CeO_2_@PQQ NPs. Furthermore, EMT6/P exhibited a 2.5-fold increase in micronuclei and a 2-fold reduction in survival fraction compared to L929. It is hypothesized that CeO_2_@PQQ NPs may exhibit selective cytotoxicity and radiosensitizing properties against EMT6/P cancer cells. The findings suggest that CeO_2_@PQQ NPs may have potential as a selective redox-active antioxidant/pro-oxidant in response to X-ray radiation.

## 1. Introduction

Exposure to ionizing radiation, an integral part of cancer radiation therapy, always leads to the development of oxidative stress. It is unfortunate that ionizing radiation affects not only cancer cells but also normal cells. In both normal cells and cancer cells, reactive oxygen species (ROS) exhibit a concentration-dependent duality. In other words, adequate concentrations of ROS are essential for maintaining redox homeostasis and for initiating cellular processes such as proliferation, growth, differentiation, and migration. On the contrary, high concentrations of ROS have cytotoxic effects, such as the activation of apoptosis and the inhibition of resistance to anticancer therapy [1]. Given the dual nature of ROS, innovative approaches to reduce or increase ROS levels have potential for cancer prevention or treatment.

In this regard, it is relevant to develop and search for new antioxidants and radioprotectors capable of neutralizing oxidative stress and ideally have selective redox activity. The goal is to reduce oxidative stress levels in normal tissues while enhancing damage to cancer cells. This presents a significant challenge for classical antioxidants, but modern nanomaterials can be used for these purposes.

Metal oxide nanoparticles (NPs) demonstrate impressive biological activity and are considered to be promising components for the creation of multifunctional antioxidants. These inorganic nanobiomaterials are characterized by a number of important physicochemical properties, such as ultrasmall particle size, high reactivity, biocompatibility, and immunogenicity, which make them suitable for biomedical applications [2,3,4,5]. CeO_2_ NPs, which represent a new class of inorganic nanozymes, have attracted special attention. The increased interest of researchers in nanoscale CeO_2_ is due to its self-regeneration, that is, the ability to repeatedly participate in redox reactions, and its multi-enzymatic activity [6,7], such as superoxide dismutase (SOD-) [8,9,10], catalase (CAT) [11,12,13], peroxidase [14,15], and other activities [16,17,18,19]. The multi-enzymatic activity of nanoscale CeO_2_ is known to be strongly pH-dependent [16,20]. This enables switching between the anti- and pro-oxidant properties of CeO_2_ NPs, which is extremely important for biomedical applications. The biological activity of CeO_2_ NPs can be affected by various factors, one of which is the surface oxidation state, which is believed to significantly affect the enzyme-mimetic activities of nanoscale CeO_2_ [21]. In this regard, the modification of CeO_2_ NPs by an antioxidant that affects the function and biogenesis of mitochondria is of particular interest. It is important to note that ROS and free radicals are associated with many diseases, including epidemiologically significant diseases [22,23,24], and lead to a secondary increases in ROS levels in irradiated cells, primarily in the mitochondria. This secondary activation is associated with damage to mtDNA and the components of the electron transport chain (ETC) complexes, which in turn leads to mitochondrial dysfunction, additional DNA damage, and the development of pathological processes. It is therefore imperative to develop new antioxidants with combined action with high catalytic activity and mitochondrial activity.

Pyrroloquinoline quinone (PQQ, vitamin B14, methoxatin) may act as such an antioxidant. PQQ is a water-soluble redox cyclic orthoquinone. PQQ has been found to be a cofactor for some bacterial alcohol dehydrogenases and a redox agent that supports mitochondrial function and biogenesis and acts as a coenzyme for several enzymes that catalyze redox reactions [25,26,27,28,29]. The uniqueness of PQQ lies in its versatility. It exhibits both reducing and oxidizing properties, similar to those of ascorbic acid [30] and riboflavin [31], respectively. PQQ has a high catalytic activity in redox reactions, more than 100 times greater than that of several vitamins with similar effects, such as ascorbic acid, menadione, isoflavonoids, and polyphenolic compounds [32,33]. PQQ and its salts have been shown to catalyze redox reactions by proton and electron transfer: for example, a single-electron scheme for the reduction of a superoxide radical involving PQQ is 10^4^ times faster than in the case of Trolox (a water-soluble analog of vitamin E) or ascorbate anion [34]. We suggest that the presence of redox-active cerium ions may further accelerate the redox reactions involving PQQ.

The current study focused on the analysis of the biological activity and radical scavenging (antioxidant) properties of new CeO_2_@PQQ NPs to investigate their effects on cytotoxicity, cell proliferation, mitochondrial membrane potential, and genoprotection upon X-ray irradiation in normal and cancer cells.

## 2. Materials and Methods

### 2.1. CeO_2_@PQQ NPs and CeO_2_ NPs Synthesis and Analysis

CeO_2_@PQQ NPs were synthesized by the precipitation method. It was investigated in detail in a previously published article [35]. For the synthesis and examination of the samples, the following reagents were used: CeCl_3_ · 7H_2_O (purity 99%, Alfa Aesar, Ward Hill, MA, USA), PQQ (purity 99%, Xi’An Horlden Bio Industries Inc., Shangluo, China), KOH (purity ≥ 98%, Lachema, Brno, Czech Republic), and NH_4_OH (25%) (Dia-M, Moscow, Russia).

Citrate-stabilized CeO_2_ NPs were selected for comparison of biological activity and provided by the IGIC RAS. Their synthesis and physicochemical properties have been described previously [36]. Nanoparticle size and zeta-potential measurements were performed using a BeNano Zeta particle size analyzer (BetterSize, Dandong, China) through dynamic light scattering and electrophoretic light scattering. UV–Vis absorption and fluorescence spectra were recorded in a quartz cuvette with a 10 mm optical path length. Absorption spectra were measured with a Shimadzu UV-1800 spectrophotometer (Shimadzu, Kyoto, Japan). The size and structural morphology of the synthesized nanoparticles were examined using a Libra 120 transmission electron microscope (TEM) (Carl Zeiss, Oberkochen, Germany) operated at 20 kV.

### 2.2. X-Ray Exposure

X-ray irradiation was used to induce oxidative stress. X-ray irradiation was performed using an RTM-15 X-ray therapeutic machine (Mosrentgen, Moscow, Russia) at a dose of 5 Gy for the NP solution and cell cultures at a dose rate of 1 Gy/min, 200 kV voltage, 37.5 cm focal length, and a 20 mA current. Cells were irradiated in T-25 flasks (SPL Lifesciences, Pocheon, Republic of Korea) in an adherent state. Solutions with NP samples and PQQ for studying catalytic activity were irradiated in Eppendorf-type test tubes (GenFollower, Shaoxing, China) A total dose of 5 Gy was delivered to cell cultures and solutions within 5 min.

### 2.3. Assessment of CeO_2_@PQQ NP Catalytic Activity

To determine the antioxidant activity of NPs (and PQQ for comparison), we analyzed the concentration of hydrogen peroxide after X-ray irradiation (5 Gy) by enhanced chemiluminescence using a luminol–4-iodophenolπperoxidase system [37]. TRIS buffer was used to maintain a constant pH (5.5; 6.8; 7.4; 8.0). A liquid scintillation counter, Beta-1 (MedApparatura, Kyiv, Ukraine), operating in the mode for counting single photons (with one photomultiplier and the coincidence scheme disengaged), was used as a highly sensitive chemiluminometer. The high sensitivity of this method allowed for the detection of hydrogen peroxide at a concentration of <1 nM. The H_2_O_2_ content was determined from the calibration dependence of the chemiluminescence on the concentration of H_2_O_2_ in the solution. The concentration of H_2_O_2_ used for calibration was determined spectrophotometrically at 240 nm with a molar absorption coefficient of 43.6 M^−1^ × cm^−1^.

### 2.4. Cell Culture

Experiments were performed using a culture of mouse fibroblasts (L929, NCTC L929) and of mouse adenocarcinoma (EMT6/P) obtained from the cryostorage of the Theranostics and Nuclear Medicine Laboratory (ITEB RAS, Pushchino, Russia). Cells were cultured in DMEM/F12 (1:1) culture medium containing 50 μg/mL penicillin, 50 μg/mL streptomycin, 10% fetal bovine serum (FBS), and 1% l-glutamine at 37 °C in a 95% humidity atmosphere containing 95% air and 5% CO_2_. Cells were seeded on 96-well and 6-well plates at a density of 25,000 cells/cm^2^. Cells from the control groups were cultured without incubation with the studied samples. For cytotoxicity studies, cells were co-incubated with CeO_2_@PQQ NPs, CeO_2_ NPs, and PQQ at concentrations of 0.1–100 μM. For other methods, data are presented only for samples at 2 μM, as this was where the most significant and pronounced effects were observed. For non-irradiated groups, cells that were not exposed to any radiation were used as the control; for irradiated groups, cells that were irradiated at a dose of 5 Gy were used as the control.

### 2.5. MTT Assay

Cell viability was analyzed using the MTT assay. This method is based on the phenomenon of the reduction of water-soluble MTT salt (3-(4,5-dimethylthiazol-2-yl)-2,5-diphenyl-tetrazolium bromide) (PanEko, Moscow, Russia) to purple water-insoluble formazan via cellular NAD(P)H-dependent oxidoreductases. After 24, 48, and 72 h of cell incubation in the presence of CeO_2_@PQQ NPs, the culture medium was replaced with an MTT solution (0.5 mg/mL) in the medium without serum. After 3 h of incubation, the medium containing MTT residue was removed, and DMSO (PanEko, Russia) was added to the wells of the plate. The plates were then placed on a plate shaker to mix the contents of the wells and dissolve the resulting formazan in DMSO. After 10 min of mixing, the absorbance of the solutions was measured at a wavelength of 540 nm using a Multiscan FC plate spectrophotometer (Thermo Fisher Scientific, Waltham, MA, USA). The absorbance values were converted into percentages of the values of the control groups, and the deviations in the samples are expressed as standard deviation (SD).

### 2.6. Live/Dead Assay

The cytotoxic effect of the CeO_2_@PQQ NPs was assessed using a Live/Dead assay. This method was used to calculate the percentage ratio of the number of cells that died during incubation with the CeO_2_@PQQ NPs to the total number of cells. Cell numbers were counted by fluorescence microscopy, and cells were labeled by staining with a combination of the fluorescent dyes: Hoechst 33342 (binds to DNA of all cells, excitation wavelength 350 nm, emission wavelength 460 nm) and propidium iodide (binds to DNA of dead cells, excitation wavelength 535 nm, emission wavelength 615 nm). After 48 and 72 h of incubation, the culture medium was replaced with a mixture of Hoechst 33342 and propidium iodide in Hanks’ buffer solution (PanEko, Russia). After 15 min of incubation with dyes, the cells were washed three times with Hanks’ buffer solution and were further analyzed using a ZOE imager (Bio-Rad, Hercules, CA, USA). ImageJ 1.54h software was used to count the number of cells. Three areas on the field were analyzed from three different microphotographs.

### 2.7. Clonogenic Assay

Cells were seeded in 6-well plates (SPL LifeScience, Republic of Korea) at a concentration of 1500 cells per well in DMEM/F12 + 10% FBS culture medium and cultured at 37 °C in an atmosphere containing 5% CO_2_. Cells were cultured in DMEM media with FBS for 24 h at 37 °C and 5% CO_2_ and then exposed to 2 μM CeO_2_@PQQ NPs, CeO_2_ NPs, and PQQ for 24 h. The medium in the wells was then replaced, and the cells were irradiated at a dose of 5 Gy. After completion of colony formation in the control group (8–9 days), cells were washed three times with phosphate-buffered saline (PBS), fixed in 4% paraformaldehyde solution (PFA) (PanReac AppliChem, Barcelona, Spain), and stained with 0.1% crystal violet (PanEko, Moscow, Russia). Cell aggregates of more than 50 cells were considered as 1 colony. Colonies were counted using ImageJ 1.54h software. To determine the survival fraction, the number of colonies formed was divided by the number of cells seeded and then normalized to the plating efficiency of the non-irradiated control samples [38].

### 2.8. Cytokinesis Block Micronucleus Assay (CBMN)

Cells were incubated with samples (CeO_2_@PQQ, CeO_2_ NPs and PQQ) at a concentration of 2 μM for 24 h. Then, the medium in the wells was replaced, and the cells were exposed to irradiation (5 Gy). Next, Cytochalasin B (2 µg/mL) (PanEko, Moscow, Russia) was added to the cells to block cytokinesis. Then, after one cell division, cells were fixed with 4% PFA (PanReac AppliChem, Spain) and stained with 4,6-diamidino-2-phenylindole (DAPI) (0.6 μg/mL) (ServiceBio, Wuhan, China). Results are expressed as the total number of micronuclei per 1000 binucleated cells. Cell proliferation was evaluated, using the nuclear division index (NDI), which indicates the average number of cell cycles. NDI was calculated using the following formula: NDI = [(1’N1) + (2’N2) + (3’N3) + (4’N4)]/1000 (N1: the number of cells with one nucleus; N2: the number of cells with two nuclei; N3: the number of cells with three nuclei; N4: the number of cells with four nuclei) [39]. Cell images were acquired using a ZOE cell imaging device (Bio-Rad, Hercules, CA, USA).

### 2.9. Mitochondrial Membrane Potential (MMP)

Mitochondrial membrane potential (MMP) was measured by staining cells with fluorescent dye tetramethylrhodamine (TMRE) (Invitrogen, Carlsbad, CA, USA) (excitation wavelength 550 nm, emission wavelength 575 nm). Cells were seeded into 96-well plates at a density of 2.5 × 104 cells/cm^2^. The cells were then incubated with CeO_2_@PQQ NPs, CeO_2_ NPs, and PQQ at 2 μM in culture medium supplemented with 10% FBS. After 24 h of incubation, cells were exposed to X-ray radiation (5 Gy) and stained with fluorescent dye. The fluorescence intensity of the dye in cells was analyzed using a Biotek Synergy H1 tablet reader (Agilent, Santa Clara, CA, USA).

### 2.10. Intracellular Glutathione (GSH)

To assess intracellular reduced glutathione (GSH) levels, cells were incubated in serum- and phenol red-free DMEM/F12 with the thiol detection reagent ThiolTracker™ Violet (10 μM/L⁻^1^) (ThermoFisher Scientific, USA) for 30 min at 37 °C. After incubation, the cells were washed twice with PBS and immediately analyzed using a Biotek Synergy H1 microplate reader (Agilent, USA).

## 3. Results

### 3.1. Characterization of CeO_2_@PQQ NPs

The synthesis of CeO_2_@PQQ NPs involved obtaining a sol of ‘bare’ CeO_2_ NPs, with-out the use of a stabilizer, forming a complex/modification of PQQ NPs and subsequent stabilization with NH_4_OH. CeO_2_@PQQ NPs were synthesized via the precipitation method [35]. The structure of the CeO_2_@PQQ NPs that were obtained is shown in Figure 1A. TEM revealed that the prepared CeO_2_@PQQ NPs had an average size of 3 nm (Figure 1B,C). The zeta potential of the CeO_2_@PQQ NPs was determined to be −38.2 mV (Figure 1D), indicating high colloidal stability. The SEM images showed that the CeO_2_@PQQ NPs were spherical in shape (Figure 1E). Spectroscopic analysis (Figure 1F) showed that the PQQ absorption peaks underwent a shift and change upon nanoparticles synthesis. This shift was particularly evident in the bathochromic shift of the 248 nm peak, which shifted to 257 nm, and the disappearance of the 275 nm peak, while the dynamic light scattering showed 55 nm as the hydrodynamic diameter of the NPs (Figure 1G).

### 3.2. Catalytic Activity of CeO_2_@PQQ NPs

The data presented in Figure 2 show that the CeO_2_@PQQ NPs were able to prevent the formation of H_2_O_2_ at various pH values in water when exposed to X-rays due to their CAT-like properties. This study was conducted immediately after the irradiation of phosphate-salt buffer solutions at pH 5.5, 6.8, 7.4, and 8.0 at a dose of 5 Gy, which led to the formation of hydrogen peroxide in concentrations of 400 nM. At pH 5.5, the CeO_2_@PQQ NPs at concentrations of 1 μM, 2 μM, and 10 μM resulted in 72%, 90%, and 100% reductions in hydrogen peroxide, respectively, compared to the irradiated control. An increase in the concentration of CeO_2_@PQQ NPs to 10 μM provided a pronounced antioxidant effect, which was expressed in a significant decrease in the concentration of hydrogen peroxide after irradiation to zero values. Interestingly, the antioxidant effect of the NPs seemed to increase as the pH value increased. At pH 8.0, all concentrations of NPs demonstrated the complete neutralization of hydrogen peroxide, indicating strong antioxidant activity.

In general, the CeO_2_ NPs demonstrated comparable hydrogen peroxide scavenging capabilities to the CeO_2_@PQQ NPs. The largest difference was observed at a 1 μM concentration (pH 5.5), where the CeO_2_ NPs showed more efficient hydrogen peroxide scavenging (72% for CeO_2_@PQQ NPs versus 81% for CeO_2_ NPs).

PQQ did not exhibit such a significant effect, and changes in pH levels did not significantly impact its activity. However, at a pH of 5.5, PQQ showed the lowest level of activity, which resulted in a peroxide reduction of about 30% relative to the irradiated control.

### 3.3. Cytotoxicity Study of CeO_2_@PQQ NPs

We conducted a study on the cytotoxicity of CeO_2_@PQQ NPs at concentrations ranging from 1 to 100 μM on mouse fibroblasts (L929) and mouse adenocarcinoma cells (EMT6/P) in order to determine non-toxic concentrations for future use in research. This study of the metabolic activity of L929 cells using the MTT assay showed that the activity of NADPH-dependent oxidoreductases after 48 h of co-incubation with CeO_2_@PQQ NPs at a concentration of 10 μM decreased by 35% relative to the control (Figure 3A), whereas those with the co-incubation of NPs at concentrations 50 μM and 100 μM with the cells were both reduced by 51%. Co-incubation for 72 h at an NP concentration of 0.1 to 2 μM had no significant effect on cell viability. However, at a CeO_2_@PQQ NP concentration of 20 μM, there was a sharp decrease in cell viability by 64%; with an increase in concentration to 50 μM and 100 μM, cell viability decreased by 87% and 81%, respectively.

According to the results of this study of the metabolic activity of EMT6/P cancer cell cultures, another effect was revealed after 48 h of co-incubation of the cancer cells with CeO_2_@PQQ NPs (Figure 3B). The metabolic activity of the cells remained at a high level up to and including a concentration of 20 μM. With an increase in the concentration above 20 μM, cells activity decreased after both 48 and 72 h. After 72 h, a gradual dose-dependent decrease in metabolic activity was shown in the range from 0.1 to 20 μM by 8% to 28%, respectively. At the same time, the concentrations of 50 and 100 μM were significantly toxic even during incubation for 48 and 72 h, reducing viability by 82 and 96%, respectively. It could also be concluded that at an NP concentration of 20 μM, the metabolic activity of both cell cultures decreased to the IC50 value for all the test samples.

The results of the Live/Dead assay demonstrated that the co-incubation of L929 with CeO_2_@PQQ NPs at a concentration of 20 μM for 48 h and 72 h led to significant increase in the proportion of dead cells to 24 and 15%, respectively (Figure 4A,B). These data are consistent with the data on metabolic activity, where a sharp decrease in viability was observed at 20 μM. At CeO_2_@PQQ NP concentrations of 50 μM and 100 μM, the number of dead cells increased to 43% and 51% at 48 h and to 57% and 53% at 72 h, respectively. Thus after 48 h and 72 h, the proportion of dead cells exceeded 50% for the 50 and 100 μM samples.

The incubation of CeO_2_@PQQ NPs with EMT6/P for 48 h showed an increase in the number of dead cells only at a concentration of 100 μM (66%) (Figure 4C). At the same time, during incubation with the NPs for 72 h (Figure 4D), cells apparently became more predisposed to the action of the NPs; here, an increase in the number of dead cells was observed, starting from the lowest concentrations. Thus, at a concentration of 0.1 µM NPs, cell mortality was 17%, and 100 µM led to 100% death of EMT6/P cells; although, for L929 at the same concentration, the maximum proportion of dead cells was 53%. Based on the data obtained, we selected CeO_2_@PQQ NPs concentrations ranging from 0.1 to 2 μM for the next stages of this study. These concentrations were chosen because they had no cytotoxic effects on any of the cell cultures.

### 3.4. Effect of CeO_2_@PQQ NPs on Antioxidant Enzyme Levels (Reduced Glutathione Measurement)

Glutathione is a key intracellular antioxidant that is capable of mitigating the oxidative stress induced by ionizing radiation. The measurement of reduced glutathione levels (GSH) in L929 cells (Figure 5A) showed that in the absence of irradiation, the CeO_2_@PQQ NPs and PQQ led to an increase in the glutathione levels (by 11% and 6%, respectively), whereas the CeO_2_ NPs had no effect. After irradiation, the GSH levels in the control group decreased slightly by 7% compared to those in the non-irradiated control. After irradiation, we observed a significant increase in glutathione in all experimental groups (8% in CeO_2_@PQQ NPs, 9% in CeO_2_ NPs, and 13% in PQQ compared to the irradiated control).

In the EMT6/P cells (Figure 5B), the GSH levels increased without irradiation in the CeO_2_@PQQ NPs (by 31%) and CeO_2_ NPs (by 22%) groups, while a decrease to 45% was observed in the PQQ group compared to the irradiated control. After irradiation, the CeO_2_@PQQ NP group showed a decrease in GSH levels by 28% compared to the irradiated control, with an even greater decrease observed in the PQQ group (by 61%). It is noteworthy that the CeO_2_ NPs resulted in a significant increase in GSH levels (by 83%), a response not observed in any other experimental group.

### 3.5. Effect of CeO_2_@PQQ NPs on Mitochondrial Membrane Potential (MMP)

It is known that X-ray radiation can lead to impaired mitochondrial function. The radiation-induced increase in the amount of ROS and calcium overload in the cytoplasm trigger the opening of the pores of the mitochondrial permeability transition and lead to a decrease in the membrane potential of mitochondria (ΔΨM), a decrease in adenosine triphosphate (ATP) production by mitochondria, and the induction of apoptosis and cell death [40]. Using fluorescent tetramethylrhodamine (TMRE) dye, which is accumulated in the mitochondria in a voltage-dependent manner, we analyzed the mitochondrial membrane potential (MMP) of the cell cultures after 72 h of incubation with CeO_2_@PQQ NPs. The CeO_2_ NPs and PQQ were used to compare the effects (Figure 6).

In the L929 cells, there were no significant differences in the MMP values compared with the non-irradiated control (Figure 6A). At the same time, in all the studied groups, the MMP was restored to the non-irradiated control values. After irradiation, the MMP decreased by 18% in the control group compared with the non-irradiated control. The CeO_2_@PQQ NPs led to an increase in MMP by 20%, CeO_2_ NPs by 10%, and PQQ by 12%, relative to the irradiated control.

The incubation of EMT6/P cells with CeO_2_@PQQ, CeO_2_ NPs and PQQ without irradiation resulted in a 24%, 46%, and 16% increase in MMP, respectively, in comparison to the control group that was not subjected to irradiation (Figure 6B). Irradiation at a dose of 5 Gy resulted in an MMP decrease by 42% in the control group. Notably, a reduction in the MMP was evident in the groups exposed to the tested samples, with a decrease of 27% and 2% observed in the CeO_2_@PQQ and CeO_2_ NP groups compared to the irradiated control, respectively.

### 3.6. Effect of CeO_2_@PQQ NPs on Colony Formation Ability

A comparative assessment of the effect CeO_2_@PQQ NPs, CeO_2_ NPs, and PQQ (2 μM) on the survival and ability to form colonies of L929 and EMT6/P cells was performed. The clonogenic assay evaluates the ability of a single cell to grow into a colony, i.e., to undergo continuous proliferation, and is often used to study the effect of ionizing radiation on the survival of cancer cells [41].

It was shown that CeO_2_@PQQ NPs led to an increase in the proliferative activity of L929, so the values of the survival fractions were 1.45 compared with the non-irradiated control (Figure 7A). When comparing the effects of the individual components of the NPs, it could be observed that the CeO_2_ NPs also significantly led to an increase in the proliferative activity and the cell survival fraction to 1.42, and PQQ to 1.24, compared with the non-irradiated control. When incubating all compounds with EMT6/P, the opposite effect was observed (Figure 7B), and all the studied samples led to a decrease in the number of cancer cell colonies (the survival fraction ranged from 0.63 to 0.69) relative to the non-irradiated control.

It was found that irradiation led to a decrease in the survival of L929 cells to 0.25 and EMT6/P to 0.68 (Figure 7A), which indicated the different radiosensitivities of the cell cultures. The co-incubation of L929 cells with all the studied compounds led to a significant increase in cell proliferative activity, and, for these groups, the survival fraction was 0.60, 0.51 and 0.38 for CeO_2_@PQQ NPs, CeO_2_ NPs, and PQQ, respectively, relative to the irradiated control. In contrast to normal cells, the co-incubation of all studied compounds with EMT6/P led to a decrease in cell survival to 0.27, 0.30, and 0.44 for CeO_2_@PQQ NPs, CeO_2_ NPs, and PQQ, respectively, relative to the irradiated control (Figure 7B).

### 3.7. Effect of CeO_2_@PQQ NPs on Genotoxicity Level

The effect of CeO_2_@PQQ NPs, CeO_2_ NPs, and PQQ on the level of micronucleus formation in L929 and EMT6/P cells was assessed using a micronucleus test with cytokinesis-block after exposure to X-ray radiation and without it, where the number of micronuclei was counted in the binucleated cells (Figure 8C). Micronuclei are extra-nuclear acentric fragments of chromosomes, or whole chromosomes, and are markers of chromosomal changes in cells [42].

It was shown that none the studied compounds led to an increase in the formation of micronuclei in L929 cells, and the CeO_2_@PQQ NPs led to a significant decrease relative to the control with and without irradiation (Figure 8A). Interesting results were obtained for the EMT6/P cancer cells, as irradiation led to an increase in the number of micronuclei in the irradiated control (Figure 8B). the co-incubation of cells with CeO_2_@PQQ NPs, CeO_2_ NPs, and PQQ increased the number of micronuclei by 20, 33 and 5, respectively, compared to the irradiated control.

The nuclear division index (NDI) was also calculated, which is a marker of cell proliferation and demonstrates the ratio of the number of cells that passed through at least one nuclear division. The calculation of the NDI of the L929 cells showed that without irradiation in all the studied groups, there were no significant differences from the non-irradiated control, whereas after irradiation, a significant stimulation of proliferative activity was shown (Figure 9A), while the most pronounced effect was observed in the group with PQQ (NDI = 1.59) and for CeO_2_@PQQ (NDI = 1.4) compared to the irradiated control (NDI = 1.25). In the case of EMT6/P cells without irradiation, significant changes were observed in the group with PQQ (NDI = 2.3) (Figure 9B). After exposure to radiation, a significant decrease in the NDI was observed in the groups with CeO_2_@PQQ NPs (NDI = 1.37) and PQQ (NDI = 1.65).

## 4. Discussion

### 4.1. Characterization and Catalytic Activity of CeO_2_@PQQ NPs

In this study, we aimed to modify CeO_2_ NPs with PQQ. Since CeO_2_ and PQQ are known for their antioxidant and prooxidant properties under various conditions, and PQQ has a broad spectrum of action on mitochondria, we aimed to enhance the redox and radioprotective/radiosensitizing properties with a new type of nanoparticle. To do this, we modified the synthesized CeO_2_ NPs via the method of precipitation with PQQ. In addition to being an antioxidant, PQQ has high catalytic activity in redox reactions and supports mitochondrial function and biogenesis. PQQ performs a radioprotective role by inhibiting oxidative stress and participating in the repair of DNA damage [33]. This study provides experimental and theoretical knowledge for the development of radioprotective clinical drugs. As noted above, PQQ and its salts catalyze redox reactions by proton and electron transfer [34]. It was important for us to assess whether the presence of redox-active cerium ion led to an acceleration of the redox reactions involving PQQ.

In our recent study, we described in detail the synthesis and examined the physicochemical properties of CeO_2_@PQQ NPs, as well as demonstrated their biocompatibility with L929 cells and their ability to act as an antioxidant system when exposed to hydrogen peroxide [35]. The results of the physicochemical properties obtained in this study (Figure 1) are consistent with those obtained previously, so CeO_2_@PQQ NPs were synthesized by the precipitation method. This synthesis technique makes it possible to obtain sol containing ultra-small CeO_2_ NPs with high catalytic and biological activities [43].

The data presented in Figure 2 show that the CeO_2_@PQQ NPs and CeO_2_ NPs were able to neutralize the hydrogen peroxide formed in water upon X-ray exposure due to their CAT-mimetic properties. Moreover, with an increase in the pH value in the phosphate-salt buffer, the catalytic properties of CeO_2_@PQQ NPs and CeO_2_ NPs increased, resulting in the complete elimination of hydrogen peroxide. Previously, we showed a similar effect of gadolinium-doped CeO_2_ NPs, where the most effective elimination of hydrogen peroxide was observed at pH 8.0 [44]. This action is likely due to the catalase-like activity that is characteristic of certain metal-based NPs [45]. CeO_2_ NPs are known to bind well to ROS, primarily hydrogen peroxide. In this case, weakly bound (adsorbed) molecules and ions are displaced from the surface of the particles [46].

As mentioned above, as a nanozyme, nanocrystalline cerium oxide mimics the activity of a number of enzymes with redox activity (SOD- and CAT-mimetic activities) [47]. CAT-mimetic properties refer to the ability of a compound to mimic catalase by decomposing hydrogen peroxide into water and oxygen, while SOD-mimetic activity refers to the ability of a compound to mimic superoxide dismutase by catalyzing the dismutation of superoxide radicals into oxygen and hydrogen peroxide [48,49,50]. The catalytic activity of these NPs is based on the unique reversible Ce^3+^/Ce^4+^ transition and the manifestation of the regenerative redox properties of the NPs. At the same time, the ratio of Ce^3+^ and Ce^4+^ on the surface of the NPs directly affects their properties: depending on the size of the NPs and the redox medium, NPs act cyclically as mimetics of SOD, associated with a higher concentration of Ce^3+^, and, as mimetics of CAT, associated with an increased content of Ce^4+^ [8,9,51]. Most importantly, this dual mechanism allows the nanoparticles to selectively target cancer cells by modulating oxidative stress, as cancer cells typically exhibit higher levels of ROS and are more sensitive to disruptions in their redox balance. In contrast, normal cells maintain a more stable redox state, allowing them to tolerate the catalytic activities of the NPs, suggesting that these properties may have potential selective effects on different cell cultures depending on their intrinsic ROS levels and redox homeostasis [52,53].

PQQ is able to carry out the redox reaction and was the most potent of the antioxidants tested [28]. It has been established that PQQ in the presence of reducing agents can initiate the production of superoxide, whereas under conditions of oxidative stress, it is able to act as an antioxidant, preventing cell damage [54]. At the same time, there are data showing that PQQ does not interact directly with H_2_O_2_ [55]. Since we observed similar effects between the CeO_2_@PQQ NP and CeO_2_ NP groups on hydrogen peroxide neutralization, we can assume that the redox activity of CeO_2_@PQQ NPs and their ability to neutralize hydrogen peroxide are largely due to the presence of cerium oxide in their composition.

### 4.2. Cytotoxicity of CeO_2_@PQQ NPs

It was found that the co-incubation of CeO_2_@PQQ NPs led to a decrease in the viability of L929 cells from a concentration of 10 µM (Figure 3A), although the number of dead cells remained insignificant. During incubation of CeO_2_@PQQ NPs with EMT6/P cancer cells, a significant decrease in cell viability was observed at concentrations from 50 µM (Figure 3B). It is known that CeO_2_ NPs and PQQ are characterized by selective cytotoxicity [56,57,58]. CeO_2_ NPs can exhibit prooxidant activity in the tumor cell microenvironment, increasing intracellular ROS levels and causing DNA damage and cell death in A-549 cancer cells, while CeO_2_ NPs do not have a toxic effect on normal cell lines such as keratinocytes and fibroblasts [59].

PQQ is primarily capable of acting as an antioxidant that neutralizes superoxide (O^2−^) and hydroxyl (HO*) radicals [60]. PQQ also affects mitochondrial function, mainly through the signaling pathways associated with mitochondrial biogenesis, such as SIRT1/PGC-1a [61]. There is evidence that PQQ can cause cancer cell death, primarily through the activation of mitochondrial-mediated apoptosis, both caspase-dependent and non-caspase-dependent, in chondrosarcoma cells [62]. At the same time, PQQ was found to have a negligible effect on normal cells, suggesting the possibility of combining it with cerium oxide to develop NPs for use in anti-cancer therapies.

### 4.3. Changes in Intracellular Antioxidant Levels (Reduced Glutathione) Under the Influence of CeO_2_@PQQ NPs and X-Ray Irradiation

One of the key factors signaling the sensitivity of tumor cells to radiation and chemotherapy is an increase or decrease in the intracellular GSH levels [63]. GSH acts as a non-enzymatic antioxidant, providing the first line of defense and playing an important role in maintaining the redox balance in cells [64]. Strategies using NPs aim to deplete GSH in cancer cells, which reduces the activity of the antioxidant systems of tumor cells and increases the development of oxidative stress [65]. It should also be noted that an acute reaction of cells to toxic effects can lead to an increase in the conversion of reduced GSH into oxidized glutathione (GSSG) during the neutralization of free radicals [66].

In this study, we found that when incubated with L929 cells, CeO_2_@PQQ NPs and PQQ led to an increase in the intracellular levels of reduced GSH, both in the absence of irradiation and after irradiation. Interesting results were obtained with EMT6/P cells, which showed an active response in the form of increased GSH levels after their co-incubation with CeO_2_@PQQ NPs and CeO_2_ NPs, but GSH levels were significantly decreased in the PQQ group. And, in the PQQ group irradiated at a dose of 5 Gy, the trend in GSH reduction was maintained. Meanwhile, with the CeO_2_@PQQ NPs, where GSH initially increased without irradiation, GSH tended to decrease with irradiation. In contrast, in the CeO_2_ NP group, GSH increased significantly with irradiation.

These results are consistent with previous findings that have shown that PQQ contributes to the depletion of GSH and another antioxidant enzyme, SOD, in chondrosarcoma SW1353 cells [67]. This suggests that PQQ may be able to inactivate the antioxidant systems in tumor cells, while increasing the level of antioxidant enzymes. For example, in a model of oxidative stress and under conditions of hyperglycemia, PQQ helps to maintain the levels of antioxidant enzymes on HepG2 cells [68] and in myocardial hypertrophy.

Our results indicate that in the EMT6/P cell culture, both an increase and a decrease in GSH levels were observed, depending on incubation with CeO_2_ NPs or CeO_2_@PQQ NPs and depending on the presence of PQQ. These findings suggest that CeO_2_ NPs and CeO_2_@PQQ NPs have different mechanisms for enhancing oxidative stress. It has been shown that cancer cells with low levels of GSH are more sensitive to radiation than normal cells [69]. However, according to our study, we found out that cells treated with CeO_2_ NPs had high GSH levels. Studies have shown that high GSH levels can lead to increased oxidative stress in BEAS-2B cells, causing a decrease in viability by 40–50% [70]. This prooxidant effect of CeO_2_ NPs was confirmed by the increased expression of genes associated with oxidative stress in cultured BEAS-2B cells [71].

### 4.4. Mitochondrial Status Under the Influence of CeO_2_@PQQ NPs and X-Ray Irradiation

The majority of GSH in cells is found in the cytosol of cells, but it is also known that maintaining the balance of GSH found in the mitochondria can also be a limiting factor in the survival of metastatic cancer cells [72]. Strategies aimed at inducing mitochondrial dysfunction have been shown to lead to the death of cancer cells [73]. When antioxidant protection is reduced and ROS are overproduced in cancer cells, oxidative damage to mitochondria can occur, leading to mitochondrial dysfunction and cell death [22]. Numerous studies have shown that damaged mitochondria can lead to the production of ROS, which can trigger apoptosis by opening the mitochondrial permeability transition pore and collapsing the MMP. These events, in turn, activate caspase-3, leading to cell death [58].

Mitochondrial potential is a measure of the functional state of mitochondria, as it reflects the movement of hydrogen ions across the inner mitochondrial membrane during the process of electron transport and oxidative phosphorylation, which drives ATP production [74,75]. This is an important indicator for understanding the health of cells and tissues. It is widely accepted that the DNA of cell nuclei represents the primary target of damage when exposed to radiation [76]. However, the mitochondria can also be considered as potential targets for ionizing radiation, given that these organelles are highly abundant in cells and have their own mitochondrial DNA [77].

In our study, we investigated the effect of irradiation on the mitochondrial potential in L929 cells (Figure 6A). We found that irradiation caused a decrease in mitochondrial potential in the irradiated control cells, but incubation with CeO_2_@PQQ NPs restored the potential to levels similar to those of the non-irradiated controls. It is worth mentioning that EMT6/P exhibited hyperpolarization of the mitochondrial membrane upon exposure to the tested samples, whereas depolarization was predominantly observed upon irradiation and incubation with the tested substances, particularly in the CeO_2_@PQQ group (Figure 6B). The changes in membrane potential are different for different cells and depend on the type of cells. Both depolarization and hyperpolarization indicate a change in mitochondrial permeability [78].

It is noteworthy that even NPs that do not initially target the mitochondria can affect these organelles. CeO_2_ NPs have been demonstrated to reduce the loss of mitochondrial potential under different types of oxidative stress modulation. Depolarization of the mitochondrial membrane may indicate oxidative damage to the mitochondria, resulting in the release of Ca^2^⁺ from the endoplasmic reticulum with the consequent increased Ca^2^⁺ uptake by the mitochondria [79]. These two important factors may contribute to mitochondria-mediated apoptosis. Previously, CeF NPs were shown to increase mitochondrial potential in MCF-7 breast carcinoma cells at a dose of 15 Gy in MCF-7 culture, which was also associated with an increase in mitochondrial size [80]. The A549 lung cancer cell line also showed an increased mitochondrial potential after irradiation, which correlated with increased oxygen uptake and ATP production [81]. With the addition of PQQ, in vitro studies have shown a decrease in oxidative stress in mouse cardiomyocytes and protection against oxidative stress in isolated liver mitochondria [27,82]. In vivo studies have shown that PQQ deficiency in mice causes the inhibition of mitochondrial function and a decrease in the number and size of the mitochondria [28].

There are also studies suggesting that PQQ can induce apoptosis of EMT6/P cancer cells. This is associated with the increased activation of caspase-dependent pathways. Different cancer cell cultures have different sensitivities to PQQ. The proposed mechanisms of action include cell cycle arrest in the G0/G1 phase; accumulation of intracellular ROS; decreased ATP levels; disruption of MMPs in combination with increased expression of activated caspase-3 [83,84,85]. Thus, we suggest that CeO_2_@PQQ NPs contribute to an increase in DNA damage in transformed EMT6/P cells both directly and through activation of mitochondrially mediated apoptotic pathways. At the same time, these NPs do not have a negative effect on normal L929 cells. In combination with PQQ, which mainly affects mitochondria, CeO_2_ NPs show potential for regulating mitochondrial functions.

### 4.5. Reproductive Cell Death and DNA Damage Under the Influence of CeO_2_@PQQ NPs and X-Ray Irradiation

The clonogenic assay measures the ability of a single cell to form a colony. Basically, the clonogenic assay enables an assessment of the differences in reproductive viability (ability of cells to produce progeny, i.e., a single cell to form a colony of 50 or more cells) between untreated control cells and cells that have undergone various treatments such as exposure to ionizing radiation or various chemical compounds (e.g., cytotoxic agents) [86]. We conducted a comparative study to assess the effect of NPs and PQQ on cell survival in the presence and absence of ionizing radiation (Figure 7).

The main cause of cell death after irradiation is due to lethal DNA damage [87]. The effectiveness of the repair of such damage actually determines the fate of the irradiated cells: maintenance/restriction of proliferation and death [88]. It is known that approximately 80% of all radiation-induced DNA damage is caused by the formation of large amounts of free radicals and ROS, which result from the radiolysis of water [89].

Our results showed that the studied NPs reduced the proliferative capacity of the transformed EMT6/P cells while not adversely affecting the viability or radiosensitivity of normal cells. This suggests that NPs specifically damage cancer cells by increasing ROS and causing DNA damage. As mentioned above, the pH-dependent redox activity of NPs may be an important factor in the development of antitumor agents, since the metabolic differences between normal and transformed cell cultures lead to pH variations in their microenvironment [90]. Such NPs are thought to be able to catalyze the decomposition of endogenous hydrogen peroxide, which is produced in significant amounts in the tumor environment, and convert it to oxygen [91]. Thus, it can be assumed that such an action of NPs can be used to increase the effectiveness of anti-tumor chemotherapy and radiation therapy.

Some of the surviving cells exposed to radiation continue to divide, despite DNA repair errors [92]. Incorrect DNA repair leads to the formation of chromosomal aberrations and cytogenetic abnormalities, such as the formation of micronuclei [93]. The inability to repair leads to the initiation of apoptotic cell death mechanisms [94].

We investigated the effect of NPs during their co-incubation with normal and cancer cells on the number of abnormal nuclei with micronuclei using the micronucleus assay with cytokinesis blocking (Figure 8 and Figure 9). The results show that CeO_2_@PQQ NPs, CeO_2_ NPs, and PQQ when co-incubated with irradiated EMT6/P cancer cells significantly increased the number of micronuclei (almost 2-fold for CeO_2_@PQQ NPs, CeO_2_ NPs) compared to the control values. For normal L929 cells, no changes were observed.

The results obtained are consistent with the data that tumor cells can lose the apoptotic mechanism. In such cases, radiosensitization may occur through the process of mitotic catastrophe, where insufficient DNA repair occurs, and the control points of the cell cycle do not function properly [94,95].

Based on the data we obtained, we can see that the NPs had a selective effect on different cell types. It is likely that CeO_2_@PQQ NPs can enhance the effects of X-ray radiation and more effectively target unstable and damaged DNA in tumor cells. In addition, we observed that the NPs exhibited more pronounced effects compared to PQQ, which gives us reason to believe that PQQ enhances the properties of CeO_2_ NPs. The literature contains some data on the action of PQQ under X-ray irradiation conditions, primarily in vivo. PQQ contributed to the survival of mice exposed to gamma radiation at a semi-lethal dose of 4 Gy and a lethal dose of 8 Gy, leading to hematopoietic recovery [96]. In contrast to CeO_2_ NPs, there are only limited studies on the effects of PQQ under different types of radiation conditions.

While the present study provides important insights into the catalytic and selective properties of the developed CeO_2_@PQQ NPs, further investigations are required to fully elucidate their molecular mechanisms of action. In particular, studies focusing on gene expression analysis will help to uncover the regulatory pathways affected by the NPs. In addition, a more detailed assessment of the ROS levels in cells using fluorescent probes will provide critical information on the dynamics of oxidative stress induced by these NPs. Furthermore, given the promising in vitro results, further in vivo studies, particularly under irradiation conditions, are warranted to validate the therapeutic potential of the NPs in complex biological systems. These additional studies will pave the way for a deeper understanding of the mechanisms involved and the wider applicability of these NPs in cancer therapy.

## 5. Conclusions

This is the first study to examine the selective redox-active capacity and biological activity of CeO_2_@PQQ NPs. Novel CeO_2_@PQQ NPs significantly pH-dependently reduced the content of hydrogen peroxide after X-ray exposure. The data demonstrated that CeO_2_@PQQ NPs exhibited selective cytotoxicity toward normal L929 cells and EMT6/P cancer cells. CeO_2_@PQQ NPs in normal L929 cells acted as a proliferator stimulator and radioprotector, maintaining high viability, reducing number of micronuclei, as well as increasing cell survival, mitochondrial membrane potential, and the cellular antioxidant level after irradiation. Meanwhile, in EMT6/P cancer cells, the CeO_2_@PQQ NPs acted as proliferation suppressor and a radiosensitizer, reducing the cell reproductive viability, increasing the micronuclei’s count, and decreasing glutathione level, leading to the radiation-induced hyperpolarization of the mitochondria. Thus, the two investigated cell cultures were subject to different effects when co-incubated with CeO_2_@PQQ NPs and irradiated, apparently thereby triggering different cell death mechanisms. In addition, we performed a comparative analysis of CeO_2_@PQQ NPs with citrate-stabilized CeO_2_PQQ and PQQ to assess the individual contributions of the components and found that, when combined, they showed a synergistic effect. The data presented here can be used as a basis for further research on nanoparticles in other cultures of normal and cancer cells in order to better understand the broader patterns between different cell types. This research will contribute to the development of quinone-modified cerium oxide nanoparticles for use as a selective antioxidant and radiosensitizer in biomedical applications.

## Figures and Tables

**Figure 1 antioxidants-13-01445-f001:**
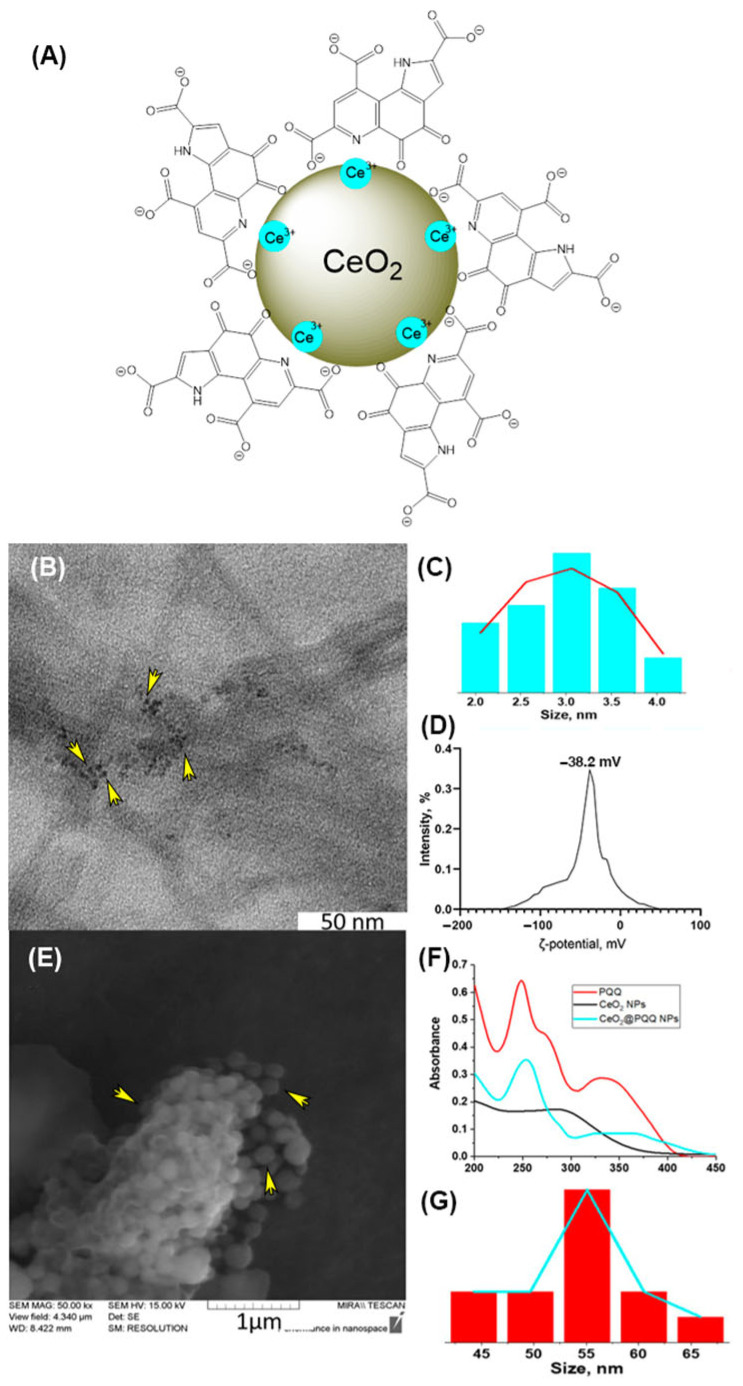
Schematic representation of CeO_2_@PQQ NPs (**A**), TEM images of CeO_2_@PQQ NPs (**B**), size distribution and Gaussian fitting (red line) of CeO_2_@PQQ NPs on the TEM image (**C**), zeta potential of CeO_2_@PQQ (**D**), SEM images of CeO2@PQQ NPs (**E**), UV–Vis absorption spectra of CeO_2_@PQQ NPs, CeO_2_ NPs and PQQ (**F**), hydrodynamic size distribution and Gaussian fitting (blue line) of CeO_2_@PQQ NPs (**G**). The arrows in the photo indicate the location of the NPs.

**Figure 2 antioxidants-13-01445-f002:**
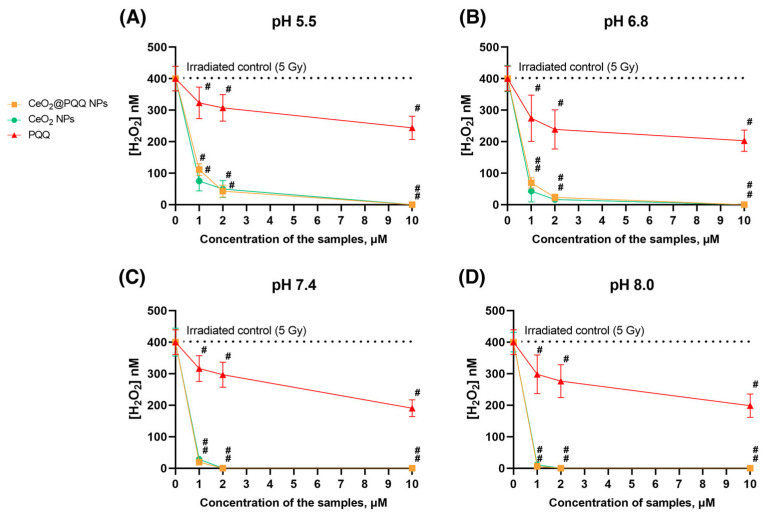
Redox activity of CeO_2_@PQQ NPs, CeO_2_ NPs, and PQQ obtained by enhanced chemiluminescence in a 4-iodophenol-luminol-horseradish peroxidase system at pH 5.5 (**A**), 6.8 (**B**), 7.4 (**C**), 8.0 (**D**). Data are shown as M ± SD and were analyzed using Welch’s *t*-test (*n* = 5) (# *p* < 0.005 vs. irradiated control, 5 Gy (400 μM)).

**Figure 3 antioxidants-13-01445-f003:**
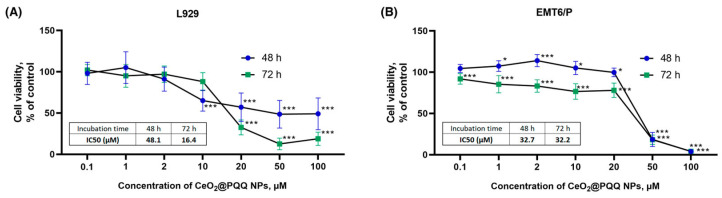
Cell viability of L929 (**A**) and EMT6/P (**B**) cells and calculation of IC50 after 48 and 72 h of incubation with CeO_2_@PQQ NPs at concentrations of 0.1–100 μM determined by MTT assay. Data are shown as M ± SD and were analyzed using Welch’s *t*-test (*n* = 5) (* *p* < 0.05, *** *p* < 0.005 vs. non-irradiated control).

**Figure 4 antioxidants-13-01445-f004:**
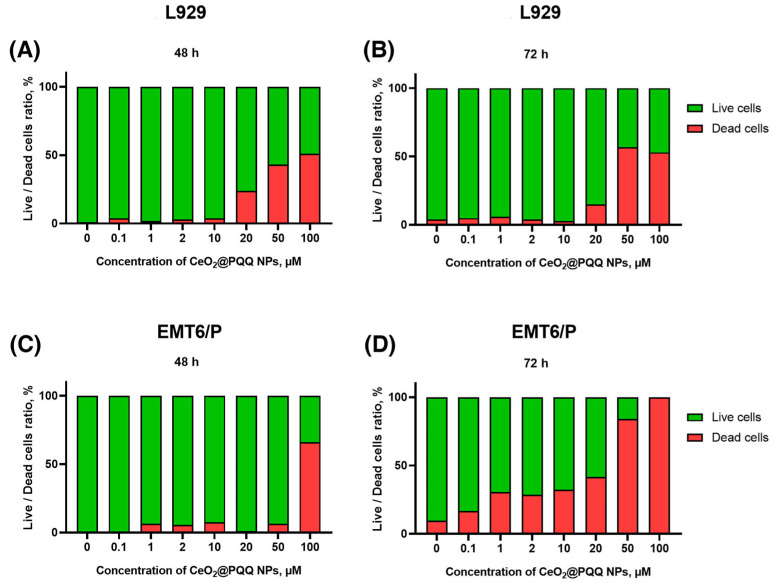
Live/Dead assay on L929 cells after 48 (**A**) and 72 (**B**) h and EMT6/P cells after 48 (**C**) and 72 (**D**) h of incubation with CeO_2_@PQQ NPs at concentrations of 0.1–100 μM.

**Figure 5 antioxidants-13-01445-f005:**
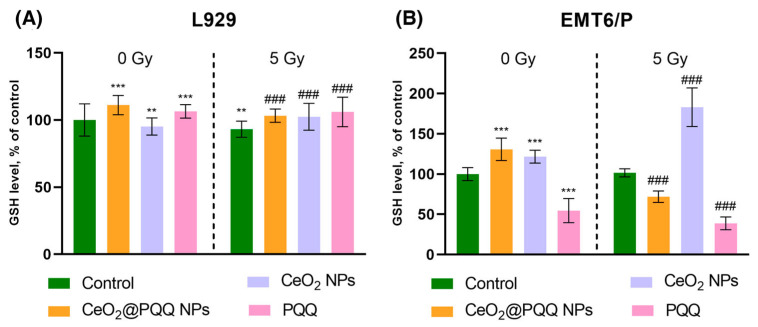
Reduced glutathione levels (GSH) in L929 (**A**) and EMT6/P (**B**) cells after incubation with CeO_2_@PQQ, CeO_2_ NPs, and PQQ at a concentration of 2 μM. Here, 0 Gy corresponds to the groups that were not exposed to radiation, while 5 Gy corresponds to the groups that were exposed to X-rays at a dose of 5 Gy. Data are shown as M ± SD and were analyzed using Welch’s *t*-test (*n* = 5) (** *p* < 0.005, *** *p* < 0.005 vs. non-irradiated control; ### *p* < 0.005 vs. irradiated control).

**Figure 6 antioxidants-13-01445-f006:**
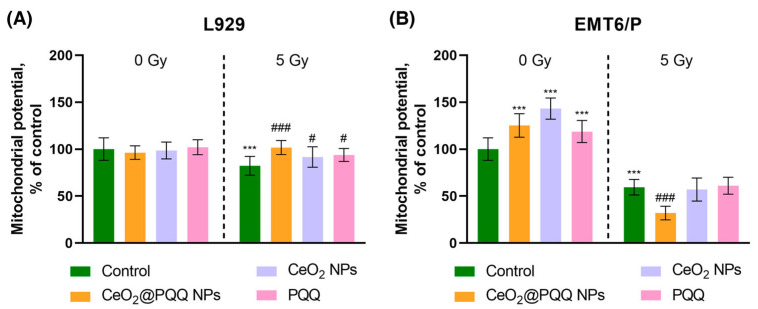
Mitochondrial membrane potential (MMP) in L929 (**A**) and EMT6/P (**B**) cells after incubation with CeO_2_@PQQ, CeO_2_ NPs, and PQQ at a concentration of 2 μM. Here, 0 Gy corresponds to the groups that were not exposed to radiation, while 5 Gy corresponds to the groups that were exposed to X-rays at a dose of 5 Gy. Data are shown as M ± SD and were analyzed using Welch’s *t*-test (*n* = 5) (*** *p* < 0.005 vs. non-irradiated control; # *p* < 0.05, ### *p* < 0.005 vs. irradiated control).

**Figure 7 antioxidants-13-01445-f007:**
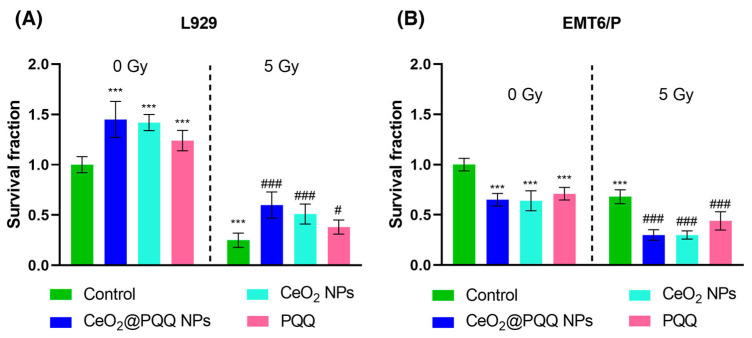
Cell survival fraction based on clonogenic assay of L929 (**A**) and EMT6/P (**B**) cells after incubation with CeO_2_@PQQ, CeO_2_ NPs, and PQQ at a concentration of 2 μM. Here, 0 Gy corresponds to the groups that were not exposed to radiation, while 5 Gy corresponds to the groups that were exposed to X-rays at a dose of 5 Gy. Data are shown as M ± SD and were analyzed using Welch’s *t*-test (*n* = 5) (*** *p* < 0.005 vs. non-irradiated control; # *p* < 0.05, ### *p* < 0.005 vs. irradiated control).

**Figure 8 antioxidants-13-01445-f008:**
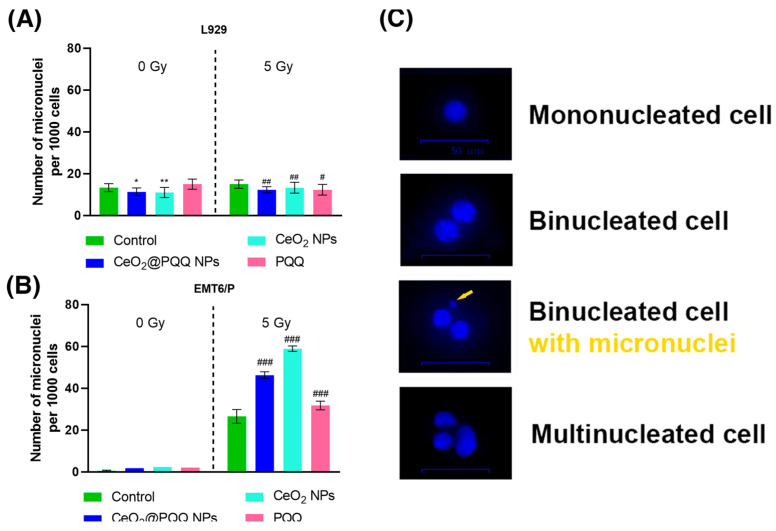
Micronucleus frequency of L929 (**A**) and EMT6/P (**B**) cells after incubation with CeO_2_@PQQ, CeO_2_ NPs, and PQQ at a concentration of 2 μM. Representative images of nuclei and micronuclei stained with DAPI are shown (the yellow arrow indicates a micronuclei) (**C**). Here, 0 Gy corresponds to the groups that were not exposed to radiation, while 5 Gy corresponds to the groups that were exposed to X-rays at a dose of 5 Gy. Data are shown as M ± SD and were analyzed using Welch’s *t*-test (*n* = 5) (* *p* < 0.05, ** *p* < 0.005 vs. non-irradiated control; # *p* < 0.05, ## *p* < 0.005, ### *p* < 0.005 vs. irradiated control).

**Figure 9 antioxidants-13-01445-f009:**
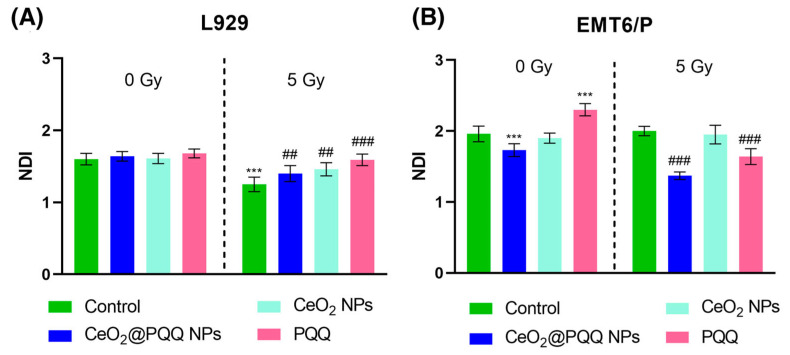
Nuclear division index (NDI) of L929 (**A**) and EMT6/P (**B**) cells after incubation with CeO_2_@PQQ NPs, CeO_2_ NPs, and PQQ at a concentration of 2 μM. Here, 0 Gy corresponds to the groups that were not exposed to radiation, while 5 Gy corresponds to the groups that were exposed to X-rays at a dose of 5 Gy. Data are shown as M ± SD and were analyzed using Welch’s *t*-test (*n* = 5) (*** *p* < 0.005 vs. non-irradiated control; ## *p* < 0.005, ### *p* < 0.005 vs. irradiated control).

## Data Availability

The data presented in this study are available in this article.

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
