# Peer review of "Novel Pyrroloquinoline Quinone-Modified Cerium Oxide Nanoparticles and Their Selective Cytotoxicity Under X-Ray Irradiation"

_antioxidants, 2024, doi:10.3390/antiox13121445_

Round 1

Reviewer 1 Report

Dear Authors,

your article entitled "Novel pyrroloquinoline quinone modified cerium oxide Nanoparticles and their selective cytotoxicity under X-ray irradiation" deals with a very topical and interesting subject, especially with regard to the protection of healthy cells from the harmful effects of radiation in cancer treatments with radiotherapy using novel nanoparticles. Therefore, this work can be considered for publication, but major revisions are necessary.

In the attached "Revisions" file, all comments are listed under:

- Methods

- Figures

- Results

- Discussion

- Minor revisions 

Author Response

Reviewer 1

Summary: Thank you very much for taking the time to review this manuscript. We greatly appreciate the comments you have provided, and we believe that your comments have helped us to substantially improve the quality of our manuscript. Please find the detailed responses below and the corresponding revisions highlighted in the re-submitted files. We have also provided the line numbers where the corresponding changes were made.

Comments 1: Since the article (ref. 35) concerning the methodical synthesis of NPs is not yet visible on the web, please briefly describe it in Methods, Section 2.1. CeO2@PQQ NPs and CeO2 NPs Synthesis and Analysis.

Response 1: Thank you for your comment regarding the description of the nanoparticle synthesis method. We are pleased to inform you that the referenced article (Ref. 35) has been published and is now available in the public domain, where the detailed methodology for synthesizing the nanoparticles is thoroughly described: https://nanojournal.ifmo.ru/en/articles-2/volume15/15-5/chemistry/paper12/

Comments 2: Please improve the quality of Figure 1A and indicate with arrows the nanoparticles. Correct the caption because there is a reversal between figure 1C and 1D.

Response 2: We have completely modified Figure 1 and marked the nanoparticles in photographs 1A and 1D with arrows. In addition, we have checked and added correct figure captions. Accordingly, the description of the results in Section 3.1. Characterization of CeO2@PQQ NPs has been corrected according to the new figure captions (Lines 217-228)

Comments 3: For easier understanding please move the legend to Figure 2 from D (pH 8.0) to A (pH 5.5). It would also have been important to include the control CeO2NPs in Figure 2 to also compare its antioxidant activity. Please provide to give more strength to your statement in Line 464-466:"we can assume that the redox activity of CeO2@PQQ NPs and their ability to neutralize hydrogen peroxide is largely due to the presence of cerium oxide in their composition"

Response 3: We have moved the legend to Figure 2 A. We have also modified the figure to include the data for the CeO2 NPs group at the same concentrations as those presented in the other groups (1, 2, and 10 μM) for comparison. Accordingly, the figure caption has also been modified. A description of the results has been added (Lines 247–250). We have also expanded the discussion to include the data obtained for CeO2 NPs (Lines 451–454, 459–463).

Comments 4: Figure 8 is missing references to A, B, and C in the caption, and Figure C is not mentioned in the text of the results

Response 4: We have corrected the caption to Figure 8 and added letter notations (Lines 412-413). We have also added a mention of Figure C in the results (Lines 396-399).

Comments 5: Please introduce in the caption of Figures 5-9 the meaning of the term 0 Gy and 5 Gy.

Response 5: We have added additional explanations to Figures 5-9 regarding the 0 Gy and 5 Gy designations.

Comments 6: Please in Line 270 replace 24 with 48 and reread and correct the sentence (269-271) "Coincubation of L929 with NPs at concentration of 20 µM for 48 h and 72 h led to significant increase in the proportion of dead cells, to 57 and 33%, respectively”. I believe it is not correct. This is not evident from the figure 4A e 4B. I see an increase of about 20% and 10 % at 48 h and 72 h respectively. Also, in the discussion (Line 468-470) you write “It was found that co-incubation of CeO2@PQQ NPs led to a decrease in the viability of L929 cells from a concentration of 10 μM (Figure 3, A), although the number of dead cells remained insignificant. In addition, in Line 285 it says “for L929 at the same concentration (100 μM), the maximum number of dead cells was 53%” It is not correct! First (Line 273-274) you say that at 100 mM dead cells increased to 51% at 48 h and 49% at 72 h. Please review the text of the results in Figure 4 and clarify this point.

Response 6: We have found discrepancies between the figures shown in the graphs and those given in the text. We have carefully checked the data and edited the text in the results section where the values were incorrect. We have made changes to Figure 3 (Lines 267-272), Figure 4 (Lines 290-297).

Comments 7: Please in Lines 373-375 these numbers and percentages, 76 (211%), 79 (219%) and 62 (172%), can be inferred from Figure 8B?

Response 7: We have corrected the values ​​for the number of micronuclei and added correct values ​​that indicate an increase in the number of micronuclei relative to the irradiated control. We have removed the percentage values ​​to avoid unnecessary confusion (Lines 404-408).

Comments 8: Please merge the introduction to the discussion in the discussion paragraph (e.g., the part about CeO2NPs) and the two paragraphs 4.1 and 4.2 turn them into one paragraph.

Response 8: We have removed the part of the Discussion that contains similar information to the text in the Introduction. We have also merged paragraphs 4.1 and 4.2 and removed the sentences describing the results of the study in detail, as already shown in the Results section.

Comments 9: Please incorporate Section 4.3 into the results at the end of Section 3.3 to justify the choice of 2 μM concentration for NPs and PQQ in the subsequent experiments in Figures 5-9.

Response 9: We have moved the concentration selection part of the Discussion to the end of the cytotoxicity results (Lines 307-310).

Comments 10: Please eliminate throughout the Discussion the repetition of the description of the results such as in Line 426-431 about Figure 1.

Response 10: We have removed the parts with repetition regarding Figure 1.

Comments 11: Please similarly thin out all other paragraphs of the discussion.

Response 11: We appreciate the important comments on the discussion section, which have helped to improve the quality of this section considerably. We have carefully revised the discussions, trying to remove repetitive parts of the text. For example, we removed from the discussion detailed descriptions of results that were already presented in the results section. We removed from the discussion parts of the text that described the mechanisms and potential applications of nanoparticles if this was mentioned in the introduction. We also removed parts that repeatedly described the same mechanisms and processes related to the action of our samples. In addition, redundant discussions were removed if they were presented several times earlier in the text (this is particularly true for the part describing the action of the mechanisms in relation to mitochondria). For example, we have merged sections 4.1 and 4.2 and substantially revised sections 4.3 and 4.4.

Comments 12: Please in line 225 correct 1 mM with 1 μM)

Response 12: Has been corrected (Line 240).

Comments 13: Please in Line 247-248 correct the sentence “While co-incubation NPs at concentrations 50 и (?) 100 μM of the cells with reduced both by 51%, respectively (Please delete)“

Response 13: Has been corrected (Lines 267-268).

Comments 14: Please check and standardize throughout the manuscript the way to indicate cerium oxidepyrroloquinoline quinone nanoparticles. CeO2@PQQ NPs or CeO2PQQ NPs?

Response 14: The correct name is CeO2@PQQ NPs. We have checked that all the names of these nanoparticles are consistent in the text and corrected where they are not (Line 24, 523).

Reviewer 2 Report

1)The paper provides detailed experimental methodologies; however, the description of the X-ray irradiation setup (Section 2.2) lacks critical details, such as the total irradiation time per sample, sample positioning, and potential heating effects. Additionally, further details on the control groups used, especially untreated cancer cells and fibroblast cells under irradiation alone, would strengthen the results.

2)The discussion could be strengthened by relating the findings more specifically to other radiosensitizers or antioxidant-based cancer therapies. Additionally, outlining future research directions, such as in vivo experiments or different cancer cell lines, would add value to the implications of this study.

3) Elaborate on the mechanism by which CeOâ‚‚@PQQ nanoparticles induce selective cytotoxicity, particularly the roles of ROS modulation, mitochondrial potential changes, and DNA damage.

4)Use additional figures or graphs to depict differences in cytotoxic effects between normal and cancer cells. Graphs showing viability, mitochondrial potential, and micronuclei counts can help readers easily interpret results.

1) While the manuscript discusses terms like “CAT-mimetic properties” and “SOD-mimetic activity,” it would be helpful to define these enzyme-mimicking terms upon first use. This would aid readers who may not be familiar with such terms. Additionally, ensure consistent abbreviations and nomenclature throughout the text to avoid confusion.

2) Although the manuscript references related studies, it could benefit from a stronger contextual foundation, especially in the catalytic activity and cytotoxicity sections. More clearly highlighting how this study advances or differs from previous work on CeO2 NPs and PQQ would enhance the significance of the findings. A brief explanation of the rationale for combining CeO2 with PQQ could also strengthen the manuscript.

3) To enhance readability, simplify the explanations of CeO2@PQQ NPs' mechanisms of action. Breaking down complex reactions, such as redox cycles and ROS neutralization, into more straightforward steps will help readers better understand these mechanisms without requiring extensive background knowledge.

4) Consider mentioning any limitations encountered in the study, such as challenges with NPs' stability or size control. Briefly discussing future research directions, such as testing NPs in vivo or exploring other combinations with CeO2, would provide a balanced perspective on the study's findings.

Author Response

Reviewer 2

Summary: Thank you very much for taking the time to review this manuscript. We greatly appreciate the comments you have provided, and we believe that your comments have helped us to substantially improve the quality of our manuscript. Please find the detailed responses below and the corresponding revisions highlighted in the re-submitted files. We have also provided the line numbers where the corresponding changes were made.

Comments 1: The paper provides detailed experimental methodologies; however, the description of the X-ray irradiation setup (Section 2.2) lacks critical details, such as the total irradiation time per sample, sample positioning, and potential heating effects. Additionally, further details on the control groups used, especially untreated cancer cells and fibroblast cells under irradiation alone, would strengthen the results.

Response 1: We have added additional information on the irradiation setup (Lines 112-116) and information on controls in the cell culture section (Lines 141-143). We also do not report on potential heating effects because, if local heating occurs at the doses studied, the effect is so small that it is generally overshadowed by the biological damage caused by ionization.

Comments 2: The discussion could be strengthened by relating the findings more specifically to other radiosensitizers or antioxidant-based cancer therapies. Additionally, outlining future research directions, such as in vivo experiments or different cancer cell lines, would add value to the implications of this study.

Response 2: Thank you for your insightful comment regarding the potential to relate the findings to other radiosensitizers or antioxidant-based cancer therapies. In the discussion section, we aimed to focus on the most relevant aspects of our results and their implications, including their alignment with current literature and potential applications. We hope that this approach maintains clarity and focus while avoiding redundancy. We concur with your observation that further studies on different normal and cancer cultures are necessary to confirm the effects we claim in this study. In addition, we believe that in vivo studies will also be a valuable contribution to the field. We have therefore incorporated these aspects into the Discussion section, where we have also outlined the limitations of the study and our future plans (Lines 655-665). Thank you for your intention to keep the discussion concise and targeted.

Comments 3: Elaborate on the mechanism by which CeOâ‚‚@PQQ nanoparticles induce selective cytotoxicity, particularly the roles of ROS modulation, mitochondrial potential changes, and DNA damage.

Response 3: We have added clarifications about the mechanisms to the discussion section. (Lines 480-486; 619-627).

Comments 4: Use additional figures or graphs to depict differences in cytotoxic effects between normal and cancer cells. Graphs showing viability, mitochondrial potential, and micronuclei counts can help readers easily interpret results.

Response 4: Thank you for your valuable suggestion to include additional figures or graphs to illustrate the differences in cytotoxic effects between normal and cancer cells. To enhance clarity and aid readers in interpreting the results, we have refined the figure 5-9 captions to better highlight the comparative effects observed in the study. We believe these updates will improve the accessibility of the data while maintaining the overall structure and focus of the manuscript. We trust that these adjustments adequately address your concern and enhance the manuscript's readability.

Comments 5: While the manuscript discusses terms like “CAT-mimetic properties” and “SOD-mimetic activity,” it would be helpful to define these enzyme-mimicking terms upon first use. This would aid readers who may not be familiar with such terms. Additionally, ensure consistent abbreviations and nomenclature throughout the text to avoid confusion.

Response 5: We have added the CAT and SOD explanations to make the text more understandable to the reader (Line 56), and we have also given a more detailed explanation of mimetic activities (Line 471-474).

Comments 6: Although the manuscript references related studies, it could benefit from a stronger contextual foundation, especially in the catalytic activity and cytotoxicity sections. More clearly highlighting how this study advances or differs from previous work on CeO2 NPs and PQQ would enhance the significance of the findings. A brief explanation of the rationale for combining CeO2 with PQQ could also strengthen the manuscript.

Response 6: Thank you for pointing this out. We have added sections on the differences between our previous study and the current study (Lines 451-454). We have also provided additional clarification on the combination of CeO2 and PQQ (Lines 438-441).

Comments 7: To enhance readability, simplify the explanations of CeO2@PQQ NPs' mechanisms of action. Breaking down complex reactions, such as redox cycles and ROS neutralization, into more straightforward steps will help readers better understand these mechanisms without requiring extensive background knowledge.

Response 7: We have tried to describe simple mechanisms, especially in relation to the effect of CeO2 and oxidation state switching (Lines 474-479). We have also simplified the parts dealing with the mechanisms of action of nanoparticles on mitochondria (Lines 594-598).

Comments 8: Consider mentioning any limitations encountered in the study, such as challenges with NPs' stability or size control. Briefly discussing future research directions, such as testing NPs in vivo or exploring other combinations with CeO2, would provide a balanced perspective on the study's findings.

 Response 8: We have added this information to the Discussion section (Lines 655-665).

Round 2

Reviewer 1 Report

Dear Authors,

the requested corrections and additions have been made, so the manuscript has been sufficiently improved and can be published but please make two small corrections indicated in the details.

Kind regards

1) Please in Line 262 after "51%" delete the word "respectively"

2) There is an error in Lines 262-264 "During co-incubation for 72 h at an NPs concentration of 0.1 to 2 μM 262 had no significant effect on cell viability. However, in the concentration of CeO2@PQQ 263 NPs of 10 μM and 20 μM, there was a sharp decrease in cell viability by 46% and 64%" 

In Figure 3A, no decrease in cell viability by 46% is observed after 72 hours of 10 μM CeO2@PQQ!! Please correct

Author Response

Reviewer 1

Summary: We would like to thank the reviewer again for his invaluable contribution to the improvement of this manuscript through important comments. Please find the detailed responses below and the corresponding revisions highlighted in the re-submitted files. We have also provided the line numbers where the corresponding changes were made.

Comments 1: Please in Line 262 after "51%" delete the word "respectively"

Response 1: The word has been deleted (Line 262).

Comments 2: There is an error in Lines 262-264 "During co-incubation for 72 h at an NPs concentration of 0.1 to 2 μM 262 had no significant effect on cell viability. However, in the concentration of CeO2@PQQ 263 NPs of 10 μM and 20 μM, there was a sharp decrease in cell viability by 46% and 64%" 

In Figure 3A, no decrease in cell viability by 46% is observed after 72 hours of 10 μM CeO2@PQQ!! Please correct

Response 2: Has been corrected (Lines 262-264)

Additional clarifications to the editor:

We would like to inform you that changes have been made to the manuscript without affecting the reviewers' comments:

1) A schematic representation of the CeO2@PQQ NP under the letter A has been added to Figure 1. The authors decided that this will help to improve the understanding of the concept regarding paragraph 3.1 in the Results sections. Accordingly, in the figure caption (Lines 224-227) and in the description of results (Lines 212-222), the notations related to the figure have been corrected.

2) The authors have added an acknowledgement to A.B. Scherbakov in the Acknowledgements section (Lines 684-685)

3) As one of the sources (ref. 35) cited by the authors was in ‘accepted for publication’ status at the time of manuscript submission and this source has now been published, the authors have provided full output data for this source, including doi (Lines 773-775)